# Impacts of burn severity on short-term postfire vegetation recovery, surface albedo, and land surface temperature in California ecoregions

**David E. Rother** [1‡]*, **Fernando De Sales**[1☉], **Doug Stow**[1☉], **Joe McFadden**[2☉]

**1** Department of Geography, San Diego State University, San Diego, CA, United States of America,
**2** Department of Geography, University of California Santa Barbara, Santa Barbara, CA, United States of America

☉ These authors contributed equally to this work.
‡ DER considered as first authorship to this work.
* drother@sdsu.edu

**Data Availability Statement:** All data for this study are publicly available. Ecoregions of California -https://www.epa.gov/eco-research/ecoregions-north-america MODIS Data Land Cover - https://

## Abstract

Wildfire burn severity has important implications for postfire vegetation recovery and boundary-layer climate. We used a collection of Moderate Resolution Imaging Spectroradiometer (MODIS) datasets to investigate the impact of burn severity (relative differenced Normalized Burn Ratio, RdNBR) on vegetation recovery (Enhanced Vegetation Index, EVI), albedo change, and land surface temperature in seven California ecoregions, including: Southern California Mountains (SCM), Southern California Coast (SCC), Central California Foothills (CCF), Klamath (K), Cascades (C), Eastern Cascades (EC), and Sierra Nevada (SN). A statewide MODIS-derived RdNBR dataset was used to analyze the impact of burn severity on the five-year postfire early-summer averages of each biophysical variable between the years 2003–2020. We found that prefire EVI values were largest, and prefire albedo and temperature were lowest in the K, C, EC, and SN ecoregions. Furthermore, the largest changes between prefire and first-year postfire biophysical response tended to occur in the moderate and high burn severity classes across all ecoregions. First-year postfire albedo decreased in the K, C, EC, and SN but increased in the SCM, SCC, and CCF ecoregions. The greatest decreases, but most rapid recovery, of EVI occurred after high severity fires in all ecoregions. After five-years post-fire, EVI and land surface temperature did not return to prefire levels in any burn severity class in any ecoregion.

## 1 Introduction

The annual burned area in California has grown since 2000, with dramatic increases in 2017, 2018, and 2020 [1]. The surge in wildfire activity in recent years is often cited as being a result of moisture deficits exacerbated by seasonal summer drought, reduced snowpack, early spring snowmelt, increased aridity, and accelerated vegetation die-off [1–4]. These wildfire risk

lpdaac.usgs.gov/products/mcd12q1v006/ Burned Area - https://lpdaac.usgs.gov/products/mcd64a1v006/ Enhanced Vegetation Index and Surface Reflectance - https://lpdaac.usgs.gov/products/mod13q1v006/ Land Surface Temperature- https://lpdaac.usgs.gov/products/myd21a2v006/ Surface Shortwave Albedo - https://lpdaac.usgs.gov/products/mcd43a3v061/.

**Funding:** The author(s) received no specific funding for this work.

**Competing interests:** The authors have declared no competing interests exist.

factors are projected to worsen under Intergovernmental Panel on Climate Change (IPCC) warming scenarios and contribute to the growing amount of total annual burned area [1, 4–6]. In California, the impacts of anthropogenic climate change on wildfire activity and wildfire related land cover change will likely vary seasonally and spatially due to the diversity of climate, topography, and vegetation distributions [1]. Also, climate change is likely to have further impacts on biophysical response by influencing the moisture conditions that can determine the rate of vegetation recovery, the species, structure, and flammability of the succeeding flora, as well as the frequency of future wildfire occurrence [6, 7].

Wildfire related land cover change on the scale of California's large burned areas alters boundary layer climate and has a significant impact on surface energy balance through changes to net radiation [8]. The immediate effect of wildfire on the land surface is the deposition of a layer of charcoal, or ash, which decreases albedo and increases sensible heat flux [8–10]. The postfire change in albedo is highly dependent on the severity of the fire and is generally short-lived, as the ash is soon dispersed by wind and rain [9, 10]. At the same time, the removal of vegetation is more often associated with a decrease in latent heat flux and an increase in sensible heating, as decreased evaporative cooling can lead to increases in land surface temperature. Land surface temperatures can rise by up to 8°C after wildfire events, with the duration of the change varying based on vegetation and ecosystem type [11–13]. Furthermore, the duration and magnitude of many of these wildfire-induced biophysical impacts are highly dependent on burn severity [10].

Burn severity refers to wildfire-induced modifications to the soil surface and vegetation conditions, and is controlled by a suite of factors including terrain slope, pre-disturbance vegetation composition, weather and climate conditions, and fuel characteristics [7, 14–17]. High severity fires are defined by complete canopy mortality and the burning of the entire top layer of soil, while low severity fires tend to burn for a shorter period of time and result in the loss of ground and understory vegetation [7, 17]. Significant alterations to the land surface associated with high severity burns change the spectral reflectance of vegetation and underlying soil, making remote sensed datasets ideal for studying their patterns. A number of studies have used satellite-derived datasets to study the effects of land surface disturbances on the recovery of vegetation in Mediterranean ecosystems [18–21]. In addition, the burn severity associated with any particular wildfire may have important implications for vegetation recovery [7, 22]. Furthermore, an ecoregion-level understanding of the biophysical response to wildfire will help land managers identify ecosystems and wildland-urban interfaces particularly at risk for the consequences of high severity fires and weakened vegetative resilience [22].

Here we investigate the impact of burn severity on vegetation recovery, albedo change, and land surface temperature in California between 2003 and 2020. In addition, we analyze the amount and distribution of burned area, EVI, surface albedo, and land surface temperature within California ecoregions, as well as the correlations between burn severity, EVI, and land surface temperature. Our metric for burn severity, a MODIS-derived relative differenced Normalized Burn Ratio (RdNBR) dataset, was used to stratify the early-summer postfire averages for three biophysical variables into three burn severity classes. Our primary research objective was to quantify the impact of burn severity on vegetation recovery, albedo, and land surface temperature across the state in the first five years after fire. Through the ecoregion-specific calculation of burned severity thresholds and their use in the analysis of EVI, land surface albedo, and temperature, we hope to help land managers foresee consequences of, and determine areas at risk for, high severity fire. In addition, five-year postfire trajectories of EVI, albedo, and temperature will improve the reliability of land surface models when simulating the impacts of wildfire on the land surface.

## 2 Materials and methods

Postfire EVI recovery, albedo change, and land surface temperature (LST), as well as all burn severity metrics, including relative differenced Normalized Burn Ratio (RdNBR), were analyzed using Moderate Resolution Imaging Spectroradiometer (MODIS) satellite data. We are aware that the coarse spatial resolution of MODIS imagery requires some sacrifice to spatial heterogeneity. However, the high temporal frequency and wide-area coverage of the MODIS data allow for a focus on the temporal patterns of vegetation recovery, albedo, and land surface temperature change.

### 2.1 California ecoregions and land cover types

The study area is California, where the U.S. Environmental Protection Agency has defined seven ecoregions used here for analysis: Southern California Mountains (SCM), Southern California Coast (SCC), Central California Foothills (CCF), Klamath (K), Cascades (C), Eastern Cascades (EC), and Sierra Nevada (SN) [23] (Fig 1). It is important to note that three ecoregions within California were excluded from the analysis. The Marine West Coast Mountains region was removed from the analysis because it experienced significantly fewer fires than the other ecoregions (~2500 total burned pixels; 3% burned area). Also, the North American Desert (i.e. southeast and northeast California) rarely experiences wildfire (~15000 total burned pixels, 2% burned area). In fact, only one large wildfire occurred in this region between 2003–2020 (i.e. the Rush Fire, Lassen County, August 2012) and so does not provide a comprehensive view of the entire ecoregion over the duration of the study period, as the study of a single fire is outside the scope of this project. Following this, the California Central Valley, a major agricultural hub in the United States, is removed from analysis due to the predominance of

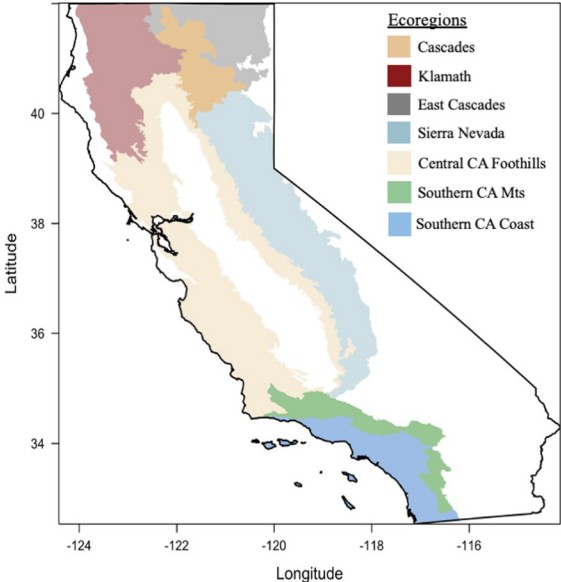

**Fig 1. Ecoregions of California, including Southern California Mountains (SCM), Southern California Coast (SCC), Central California Foothills (CCF), Klamath (K), Cascades (C), East Cascades (EC), and Sierra Nevada (SN).** Areas in white represent coastal zones, Central Valley agriculture and desert areas. The coastal zone in northwest California and the desert zone in the south and northeast are removed from analysis due to rare occurrence of wildfire. The Central Valley was removed from the analysis due to the predominance of crop land cover, irrigated agriculture, and the high frequency of low severity fires that are filtered out after the application of burn severity thresholds. Ecoregions of California (Griffith, 2016): https://doi.org/10.3133/ofr20161021.

crop land cover types and irrigated agriculture that does not experience vegetation recovery in a comparable way to other land cover types. Additionally, the Central Valley experiences a high frequency of small fires of low severity that are filtered out after the application of burn severity thresholds.

We used the MODIS Version 6 MCD12Q1 500 m Land Cover Type to analyze the pre- and postfire characteristics of the land surface [24, 25]. This product is derived from supervised classifications of reflectance data and provides 17 global land cover types, including needleleaf forest, broadleaf forest, closed shrublands, woody savanna, grasslands, and others [26]. We calculated the percentage of each vegetation type that remained unburned for the duration of the study period, as well as the mean early-summer EVI for five predominant land cover types (Table 1).

Overall, the prevailing land cover type within the unburned areas of the Klamath ecoregion is needleleaf forest (63%); the Eastern Cascades ecoregion is mostly grassland (78%); the Cascades are most substantially needleleaf forest (37%) and closed shrublands (46%); the Central California Foothills ecoregion is mostly grasslands (51%) (Table 1). For our purposes of land cover analysis, the closed shrubland category includes both closed shrublands and woody savanna vegetation types.

## 2.2 Fire history

The location and approximate date-of-burn information for California wildfires that occurred between January 2003 and December 2020 was obtained from the MODIS MCD64A1 product [27]. Additionally, this burned area product was also used to calculate the total annual burned area within the seven ecoregions of California for the years 2003–2020. MODIS identifies burned areas by measuring changes in surface reflectance [28]. The majority of land surface

**Table 1. Land cover characteristics for the unburned regions of each ecoregion averaged for 2003–2020.**

| | Southern California Mountains | Southern California Coast | Central California Foothills | Klamath Mountains | Cascades | Eastern Cascades | Sierra Nevada |
|---|---|---|---|---|---|---|---|
| Total Land (total 500 m pixels) | 89263 | 117026 | 446343 | 201138 | 84997 | 107530 | 305702 |
| Burned Area (%)[a] | 46 | 25 | 16 | 31 | 6 | 7 | 17 |
| Nonburned Areas (number of 500 m pixels) | 47861 | 88028 | 376989 | 138994 | 79548 | 99845 | 253509 |
| *Percentage of Land Cover Types (%)* | | | | | | | |
| Needleleaf | 3 | 0 | 3 | 63 | 37 | 2 | 21 |
| Closed Shrublands[b] | 14 | 8 | 13 | 23 | 46 | 6 | 21 |
| Open Shrublands | 20 | 4 | 3 | 0 | 0 | 0 | 5 |
| Savanna | 28 | 12 | 17 | 3 | 9 | 8 | 21 |
| Grasslands | 33 | 23 | 51 | 6 | 6 | 78 | 27 |
| *Mean Early-Summer EVI[c]* | | | | | | | |
| Needleleaf | 0.36 | 0.32 | 0.42 | 0.42 | 0.36 | 0.32 | 0.39 |
| Closed Shrublands | 0.24 | 0.22 | 0.3 | 0.34 | 0.29 | 0.24 | 0.33 |
| Open Shrublands | 0.18 | 0.16 | 0.18 | 0.28 | 0.17 | 0.14 | 0.15 |
| Savanna | 0.24 | 0.27 | 0.27 | 0.31 | 0.3 | 0.26 | 0.27 |
| Grasslands | 0.23 | 0.18 | 0.19 | 0.29 | 0.23 | 0.19 | 0.23 |

[a]Percent of all land cover types that burned within each ecoregion between 2003 and 2020.

[b]Closed shrublands includes closed shrublands and woody savanna.

[c]Mean early-summer (Julian day 177–224) EVI in unburned regions between 2003 and 2020.

areas that burn each year are a consequence of large wildfire events, and MODIS pixels classified as burned more than once within the study period were not included in the calculation of total burn pixels.

## 2.3 MODIS enhanced vegetation index and albedo

To analyze the presence of vegetation on the land surface, we used the MOD13Q1 Version 6 Enhanced Vegetation Index (EVI) 16-day, 250 m resolution product. MOD13Q1 can discriminate between canopy and canopy background and has an improved atmospheric correction for cloud and aerosol contamination [29, 30]. Here we use EVI as a measure of vegetation productivity and refer to vegetation recovery synonymously with EVI recovery to describe the trajectory of postfire EVI towards prefire values. Mean early-summer EVI was calculated for individual vegetation types within each ecoregion (Table 1). Mean EVI was highest for all vegetation types within the Klamath region. Overall, the SCM and the EC had the lowest average mean early-summer EVI. In addition, needleleaf forest had a higher mean early-summer EVI within each ecoregion than any other vegetation type.

For the analysis of land surface albedo, we used the 500 m resolution MCD43A3 Version 6 White-Sky shortwave albedo model product at a 3-day temporal resolution [31]. MODIS shortwave albedo (0.3–5.0 $\mu$m) has been successfully used to measure albedo response to wildfire in previous studies [9, 30, 32–35].

## 2.4 MODIS land surface temperature

Collection 6 MODIS MYD21A2 Land Surface Temperature (LST) data were used for this study to calculate five-year postfire recovery patterns within pixels of varying levels of burn severity. MYD21 is an 8-day 1-km product that is derived through a Temperature/Emissivity Separation technique using thermal infrared bands 29 (8.4–8.7 $\mu$m), 31 (10.78–11.28 $\mu$m), and 32 (11.7–12.27 $\mu$m), as well as land surface temperature [36]. In addition, a Water Vapor Scaling correction is applied to account for biases during warm and humid weather. MYD21 has been evaluated against ground-based measurements and is shown to have a bias of -0.2˚C [37–40].

## 2.5 Burn severity

We calculated Normalized Burn Ratio (NBR, Eq 1) using MOD13Q1 surface reflectance band 2 (841–876 nm) and band 7 (2105–2155 nm). NBR is a spectral index that has been widely used to analyze the post-disturbance effect of wildfire on the land surface [30, 41–46], and studies evaluating NBR against on-the-ground estimates of burn severity have found strong correlations [7, 17, 47–49]. NBR uses differences in reflectance between the near infrared and shortwave infrared wavelengths, normalized by the sum of the two bands, to measure changes to the land surface after wildfire events, such as deposition of char and ash, removal of vegetation, decreased moisture content, and exposed soil [7, 44, 50]. We calculated an early-summer NBR (June 26 –August 12) for California for each year within the time period January 2002 through December 2020.

$$NBR = \frac{((Band\ 2) - (Band\ 7))}{((Band\ 2) + (Band\ 7))} \tag{1}$$

Healthy vegetation displays strong reflectance in the near infrared band (band 2) and low reflectance in the shortwave (band 7), while recently burned areas show the opposite [17]. Therefore, low NBR values indicate recently burned areas (or low vegetation), while high NBR

values indicate healthy vegetation. Prefire NBR was calculated as the average NBR within burned area perimeters in the year before the fire, while postfire NBR was calculated by measuring average NBR in the same burn perimeter the year after the fire.

We also calculated the differenced Normalized Burn Ratio (dNBR) by subtracting average postfire NBR from average prefire NBR (dNBR = prefireNBR—postfireNBR). dNBR time series were generated for the early-summer period for each year from January 2002 –December 2021. While dNBR utilizes pre- and postfire imagery to calculate an absolute change, the relative differenced Normalized Burn Ratio (RdNBR) measures burn severity relative to prefire surface reflectance [51], and is calculated as:

$$RdNBR = \frac{dNBR}{\sqrt{|prefireNBR|}} \tag{2}$$

RdNBR assesses changes in near infrared and shortwave infrared radiation in the context of post-disturbance variations in vegetation and soil moisture [30]. RdNBR has been shown to be more robust when comparing fires across landscapes, and more accurately differentiates levels of burn severity within heterogeneous landscapes [45, 51]. Values typically range from –1.5 to +1.5, with positive numbers indicating varying degrees of burn severity (when postfire NBR is negative), while negative values indicate varying levels of vegetation recovery (when postfire NBR is positive). Burn severity thresholds for low, moderate, and high severity were derived for each ecoregion from the cumulative distribution of RdNBR for 20%-45%, 45%-75%, and >75% percentiles. These percentile groups span a different range of RdNBR values for each ecoregion, and thus are more dynamically suited to regions with varying prefire surface reflectance, which allows for a more standardized and informed classification of burn severity across landscapes compared to a single set of thresholds applied to all ecoregions. It is important to note that burned pixels with RdNBR values below the 25th percentile were considered recovered, or unchanged, and are not included in this study, as the surface reflectance values in the postfire image returned to roughly that of the prefire image after one year.

Statistical testing involved calculating the correlation coefficient for pre- and postfire NBR with RdNBR, as well as ΔEVI and ΔLST with RdNBR (change refers to the difference between the pre- and first year postfire images unless otherwise specified). Pearson correlation values were found using cortest() function in the *R* programming software. In order to obtain the correlations for each ecoregion, the pre- and postfire early-summer average of NBR, RdNBR, EVI, and LST was calculated within each year's burned area. The temporal correlation, using one average for each variable for 2002–2021 was then obtained for the pairs listed above. This correlational analysis provided a useful metric with which to analyze the dominant controlling factor in the calculation of RdNBR (pre- or postfire NBR). It was also a useful tool for understanding the relationship between postfire biophysical response and burn severity. In addition, by comparing the biophysical changes that occur between the first-year pre-fire and the first-year post-fire (i.e. differenced indices), we could quantify the largest changes between a burned and unburned land surface.

### 2.6 Five-year postfire averages

We focused on the temporal dimension of burn severity's impact on biophysical variables by analyzing the early-summer averages for EVI, shortwave albedo, and land surface temperature within seven ecoregions and three burn severity classifications during the first five years after fire. Burned areas were considered for the years 2003–2020, which allowed for the inclusion of one year pre-fire (2002) and one year post-fire (2021) at the beginning and end of the time series (i.e. five years of postfire data were not required for a given year of burned area data to

be included in the analysis). Early-summer averages were obtained for each biophysical variable by calculating their mean value within annual eco-region specific burned areas between approximately June 26 and August 12 in each year from 2002 to 2021. We chose early-summer because California's vegetation tends to peak annually during this season and this time period represents the beginning of the state's fire season. A single early-summer average from each year allows for the efficient calculation of RdNBR across multiple years, as well as a clear indication of the recovery pattern over five years post-fire. For each variable and burn severity classification we plotted postfire response for the five years following the wildfire. In addition, the pre-fire early-summer average was calculated for each variable. We understand that by calculating burn severity with surface reflectance imagery one-year post-fire, we incorporate an entire year of postfire recovery, which may vary within ecoregions. However, the high temporal frequency of the sampling in the MODIS datasets, as well as the inclusion of seven ecoregions with different predominant vegetation types and climates, provides important detail to this temporal study of biophysical response.

## 3 Results

### 3.1 Burned area and burn severity analysis

The spatial distribution of wildfires indicates that many of the large fires that occurred in the northern California ecoregions burned in the summer months of July, August, and September (Fig 2A). However, in southern California, fires burned predominantly in August, September, and October. In fact, the Thomas Fire, which burned in Santa Barbara and Ventura counties of southern California in 2017 was the only large wildfire to burn in December. Burn severity maps indicated that average RdNBR in the SCC and CCF ecoregions is predominately categorized as low severity, while the K and SCM experienced more fires of moderate and high severity (Fig 2B).

Total burned area for the entire study area during the 2003–2020 study period was 66807 $km^2$, which is ~20% of the study area. The average early-summer RdNBR falls within the low

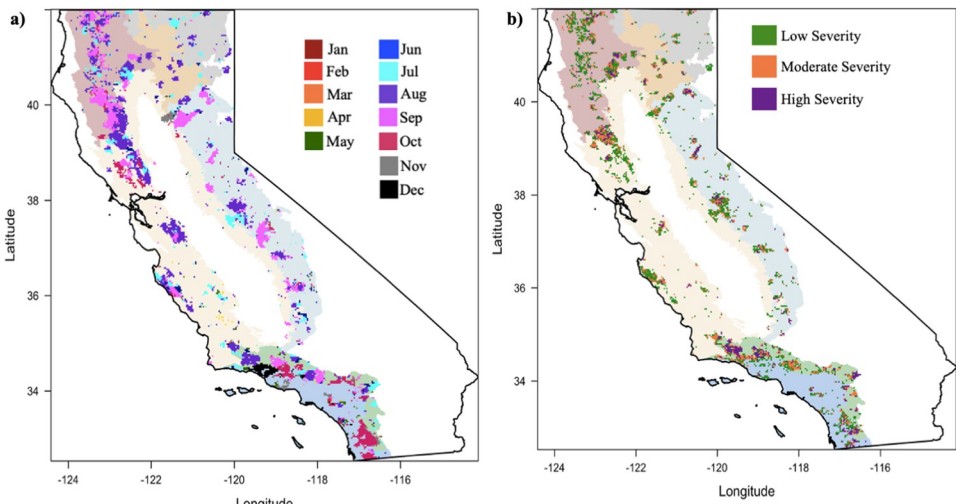

**Fig 2.** All burned areas by month for the time period of January 2003 –December 2020 (a) and all burned area by burn severity class for the time period of January 2003 –December 2020 (b). The seven ecoregions are larger polygons represented in low saturated colors (as in Fig 1). For panel (a), a binary burned-unburned mask was generated for each month of the MCD64A1 Monthly Burned Area product and the sum for each month calculated. For panel (b), RdNBR was derived from MOD13Q1 surface reflectance data, burn severity thresholds were applied, and the spatial distribution of burn severity plotted (see Methods).

burn severity range for each respective ecoregion (Table 2). However, the K, C, and SN ecoregions show the highest prefire NBR values, indicating larger amounts of prefire vegetation (Table 2).

The CCF ecoregion recorded the most burned area (17339 $km^2$), while the Cascades ecoregion recorded the least (1362 $km^2$) (Table 2). Roughly 17600 $km^2$ burned in the southern California ecoregions of SCM and SCC combined, while over 30000 $km^2$ burned in the northern California ecoregions of K, C, EC, and SN (Table 2). Warm and dry late spring and summertime meteorological conditions drive northern California fires in the months of June through August, while low levels of precipitation (4.7 mm $mo^{-1}$) and high summertime temperatures (~27°C on average), along with Santa Ana winds, drive southern California fires in August through October [17, 52, 53].

California ecoregion specific total annual burned area was plotted for 2003–2020 (Fig 3). Total burned area, including all seven ecoregions, was high in 2008, 2017, 2018, and 2020 when 6382 $km^2$, 6710 $km^2$, 9165 $km^2$, and the remarkable 22764 $km^2$ of land area were burned, respectively.

In 2008 and 2018, the most land area burned within the Klamath ecoregion; however, in 2017, more land burned in the CCF, SCM, and SCC ecoregions compared to the northern California regions. More land burned in SCM, SCC, and CCF ecoregions than in the K, C, EC, and Sierra Nevada ecoregions in eleven of the eighteen years of the study period, however, between 2012 and 2015 more area burned in K, C, EC, and Sierra Nevada. The CCF saw a steady increase in total annual burned area between 2015 and 2018, a decrease in 2019, and a dramatic increase in 2020. In fact, 2018 and 2020 each broke the previous record for annual burned area, with 2020 totals of 6904 $km^2$, 6402 $km^2$, and 5192 $km^2$ of burned area in the CCF, K, and SN ecoregions respectively (total of 18498 $km^2$ combined in a single year).

**Table 2. Burned area characteristics for each ecoregion averaged for the period 2003–2020.**

|  | Southern California Mountains | Southern California Coast | Central California Foothills | Klamath Mountains | Cascades | Eastern Cascades | Sierra Nevada |
|---|---|---|---|---|---|---|---|
| Burned Area (square kilometers) | 10351 | 7250 | 17339 | 15536 | 1362 | 1921 | 13048 |
| Number of 500 m pixels | 41402 | 28998 | 69356 | 62144 | 5449 | 7685 | 52193 |
| Summer EVI (prefire)[a] | 0.24 (±0.3) | 0.19 (±0.02) | 0.25 (±0.05) | 0.4 (±0.05) | 0.29 (±0.07) | 0.27 (±0.13) | 0.31 (±0.04) |
| Summer EVI change (first year after fire)[b] | -0.07 (±0.04) | -0.02 (±0.03) | -0.06 (±0.03) | -0.15 (±0.04) | -0.1 (±0.05) | -0.04 (±0.04) | -0.1 (±0.04) |
| Summer RdNBR | 0.46 (±0.21) | 0.29 (±0.55) | 0.34 (±0.14) | 0.44 (±0.11) | 0.32 (±0.34) | 0.24 (±0.25) | 0.47 (±0.14) |
| Summer dNBR | 0.26 (±0.12) | 0.07 (±0.08) | 0.17 (±0.08) | 0.33 (±0.1) | 0.27 (±0.16) | 0.1 (±0.1) | 0.3 (±0.12) |
| Summer NBR (prefire) | 0.21 (±0.08) | 0.05 (±0.06) | 0.23 (±0.1) | 0.58 (±0.11) | 0.37 (±0.14) | 0.22 (±0.19) | 0.4 (±0.11) |
| Summer NBR (first year after fire) | -0.01 (±0.06) | -0.01 (±0.07) | 0.07 (±0.07) | 0.24 (±0.1) | 0.14 (±0.12) | 0.12 (±0.21) | 0.1 (±0.07) |
| Summer Albedo (prefire) | 0.13 (±0.02) | 0.14 (±0.01) | 0.15 (±0.02) | 0.11 (±0.01) | 0.11 (±0.01) | 0.13 (±0.02) | 0.11 (±0.01) |
| Summer Albedo (first year after fire) | 0.15 (±0.02) | 0.15 (±0.01) | 0.15 (±0.02) | 0.1 (±0.01) | 0.11 (±0.02) | 0.13 (±0.02) | 0.11 (±0.01) |
| Summer LST (prefire) | 314.6 (±4.07) | 316.68 (±2.26) | 316.68 (±3.77) | 305.4 (±2.97) | 308.42 (±3.09) | 314.47 (±5.48) | 308.24 (±2.54) |
| Summer LST (first year after fire) | 320.33 (±2.73) | 3.19 (±2.64) | 320.65 (±2.97) | 309.26 (±2.86) | 313.11 (±4.52) | 317.44 (±6.02) | 313.26 (±3.24) |

Includes total number of 500 m pixels that registered as burned area, burned area (sq. km), relative differenced Normalized Burn Ratio (RdNBR), and the pre- and postfire response of Normalized Burn Ratio (NBR), Enhanced Vegetation Index (EVI), surface shortwave albedo, and land surface temperature (LST). Mean values are calculated within each year's burned area. Standard deviation of the ecoregion means is shown in parentheses.

[a] Early-summer averages are calculated from Julian Day 177–224

[b] Change refers to the difference between the prefire value and the first-year post-fire

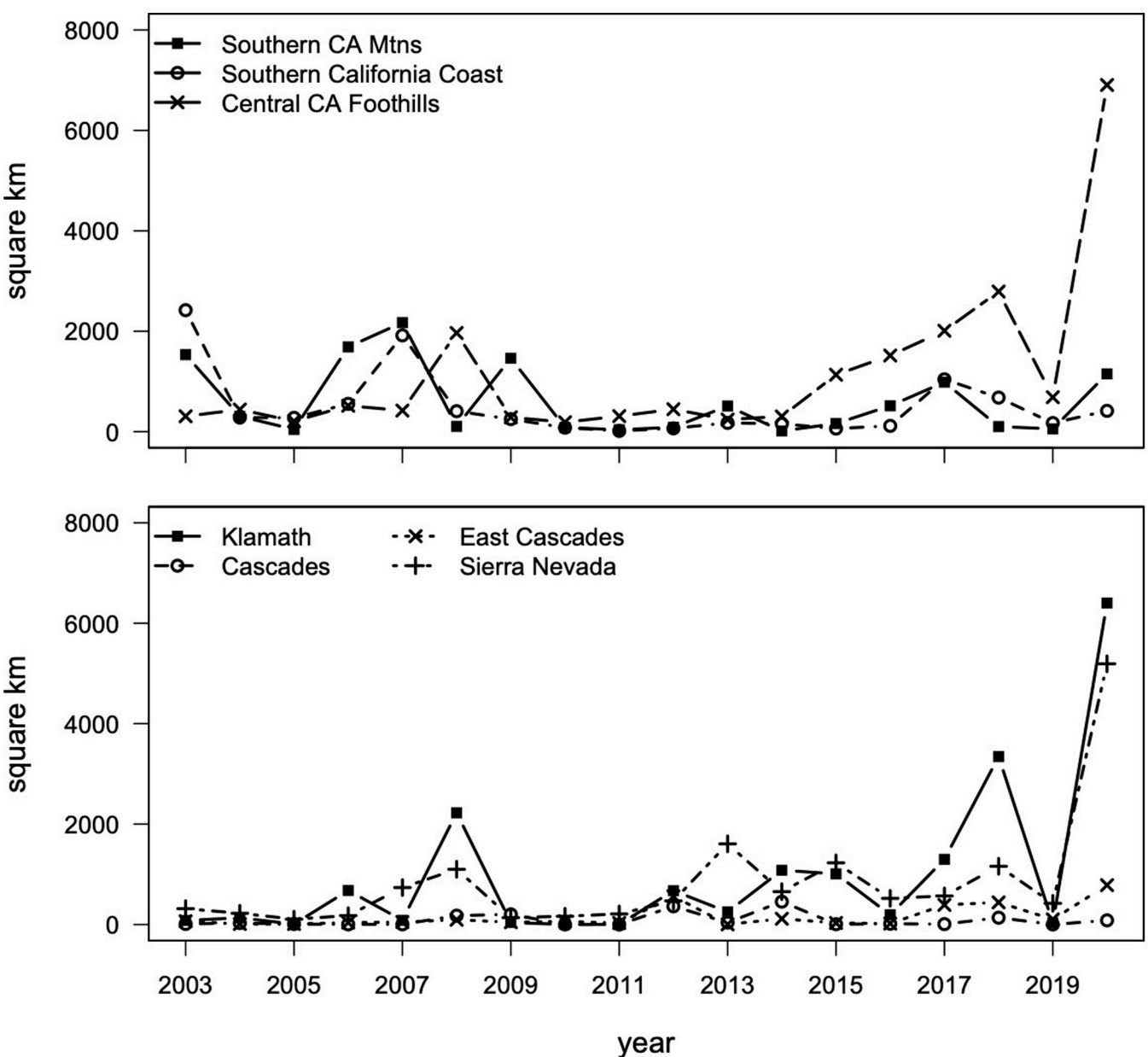

**Fig 3. California ecoregion specific total annual burned area for 2003–2020 based on MODIS MCD64A1 approximate date-of-burn product.** Ecoregions are split into roughly southern (top) and northern (bottom) California. Total burned area each year represented in square kilometers.

RdNBR frequency distributions were plotted for each ecoregion and for several dominant land cover types found throughout California, including needleleaf forest, closed and open shrubland, savanna, and grassland (Fig 4). RdNBR derived burn severity threshold ranges were recorded below (Table 3). The lower limit of the low burn severity threshold ranged from 0.13 in the SCC to 0.44 in the SCM. The lower limit of the moderate burned severity threshold ranged from 0.33 in the SCC to 0.63 in the SCM, while the lower limit of the high burn severity threshold ranged from 0.65 in the SCC to 0.82 in the SCM (Table 3). The upper limit of +1.5 for the high burn severity threshold was chosen because RdNBR values start to reach their

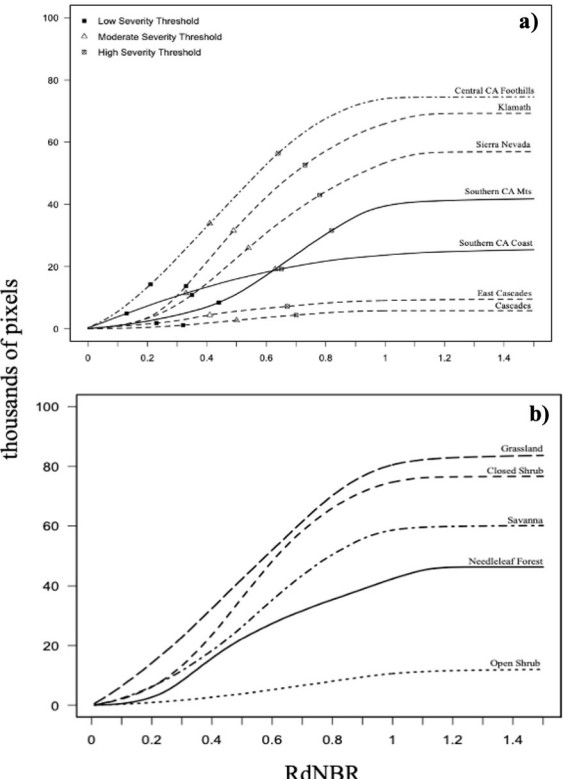

**Fig 4. Histograms for RdNBR averaged for the entire study period of 2003–2020.** Symbols mark the lower limits of low, moderate, and high burn severity class respectively; (a) RdNBR by ecoregion, and (b) RdNBR by individual vegetation type. Burn severity thresholds for each ecoregion were calculated from the cumulative frequency distribution of RdNBR pixel values (see Methods).

asymptote in this range (see Fig 4), and values above this limit are likely caused by misregistration, clouds, or other non-land cover related anomalies [15].

The cumulative frequency distribution of RdNBR values within ecoregions indicates that the SCC, CCF, and EC have the greatest frequency of RdNBR values <0.4, thus the lower limits to the low burn severity thresholds are lower than in other ecoregions (Fig 4 and Table 3). Conversely, the SCM, SN, and K ecoregions have the lowest frequency of RdNBR values <0.4, thus the lower limit to these region's low burn severity thresholds are comparably higher. According to the cumulative distribution, a greater number of pixels of RdNBR <0.4 will decrease the limit of the low burn severity class (i.e. SCC), while a distribution skewed towards

**Table 3. RdNBR low, moderate, and high burn severity thresholds for each ecoregion derived from RdNBR cumulative distributions.**

| RdNBR Thresholds | Low | Moderate | High |
|---|---|---|---|
| So. CA Mountains | 0.44–0.62 | 0.63–0.81 | 0.82–1.5 |
| So. CA Coast | 0.13–0.32 | 0.33–0.64 | 0.65–1.5 |
| Central CA Foothills | 0.21–0.4 | 0.41–0.63 | 0.64–1.5 |
| Klamath | 0.33–0.48 | 0.49–0.72 | 0.73–1.5 |
| Cascades | 0.32–0.49 | 0.5–0.69 | 0.7–1.5 |
| East Cascades | 0.23–0.4 | 0.41–0.66 | 0.67–1.5 |
| Sierra Nevada | 0.35–0.53 | 0.54–0.77 | 0.78–1.5 |

higher RdNBR values will increase the lower limit (i.e. SCM). So, in this way, regardless of the variation in size of each ecoregion and the total number of burned pixels, it is the distribution of RdNBR values across the burn severity spectrum that determines each threshold.

Similar analysis of the RdNBR histogram by vegetation type indicates that grasslands burn at low severity more than other ecoregions, while needleleaf forest and open shrub experience a greater frequency of RdNBR >0.8 (Fig 4B). While closed shrub burns at a greater frequency compared to savanna, their distributions of RdNBR values are similar until ~0.3, at which point the closed shrub distribution increases more rapidly through the moderate burn severity (0.4–0.6) range compared to savanna. Overall, grassland and closed shrub yielded the greatest number of pixels affected by some level of burn severity one-year post-fire (Fig 4B).

Correlations between average early-summer burn severity (RdNBR) and postfire NBR were all negative, with the strongest correlations in the SCM (-0.89) and SCC (-0.76) (Table 4). In addition, early-summer RdNBR and ΔEVI were highly negatively correlated for each ecoregion, indicating that high burn severity is associated with wildfire-induced vegetation removal (i.e. more vegetation burned by wildfire). Strong correlations between RdNBR and ΔEVI are expected because both indices are sensitive to infrared reflectance [15, 30].

In addition, correlations between ΔLST and burn severity are all positive and significant (except for SCC), ranging from 0.69 (CCF) to 0.89 (SN) (Table 4). This positive relationship implies that larger increases in postfire land surface temperatures are associated with higher burn severity. These increases in postfire temperature associated with high severity fires have severe implications for ecosystem health and vegetation recovery.

## 3.2 Vegetation recovery

The highest prefire EVI values and the greatest decrease of EVI after one-year post-fire, regardless of burn severity class, were found in the K, C, and SN ecoregions (Fig 5D, 5E and 5G). Early-summer EVI decreased in the year following wildfire due to vegetation removal, and high severity fires were associated with the largest decrease and lowest value of EVI one year later (Fig 5). Decreases in EVI for low severity fires after one year ranged from 0.02 in the SCC to 0.14 in the K, while decreases in EVI after high severity fires ranged from 0.06 in the SCC to 0.25 in the K (Fig 5).

The rate at which EVI recovered towards the prefire value was different for each ecoregion. EVI did not recover to prefire levels after five years post-fire for any burn severity class in any ecoregion. However, during the first two years post-fire, EVI recovery occurred more rapidly after high severity burns in all ecoregions. Overall, the SCC exhibited the least amount of

**Table 4. Pearson temporal correlations (r values) between average early-summer burn severity (RdNBR) and burned area averages of six biophysical variables within each of the seven ecoregions of California.**

|  | Southern California Mountains | Southern California Coast | Central California Foothills | Klamath Mountains | Cascades | Eastern Cascades | Sierra Nevada |
|---|---|---|---|---|---|---|---|
| Prefire NBR | **0.77** | 0.04 | 0.42 | 0.28 | **0.71** | -0.12 | **0.58** |
| Postfire NBR | **-0.89** | **-0.76** | **-0.52** | **-0.61** | -0.43 | **-0.55** | **-0.62** |
| Prefire EVI | **0.77** | 0.18 | 0.36 | 0.27 | **0.6** | -0.29 | **0.47** |
| ΔEVI [a] | **-0.9** | **-0.57** | **-0.78** | **-0.83** | **-0.85** | **-0.82** | **-0.86** |
| Prefire LST | **-0.69** | -0.35 | -0.26 | -0.28 | 0.04 | -0.12 | -0.36 |
| ΔLST | **0.87** | 0.35 | **0.69** | **0.77** | **0.67** | **0.84** | **0.89** |

Bold numbers represent correlations significant at 95% confidence level (p<0.05).

[a] Change refers to the difference between the pre-fire value and the first-year post-fire

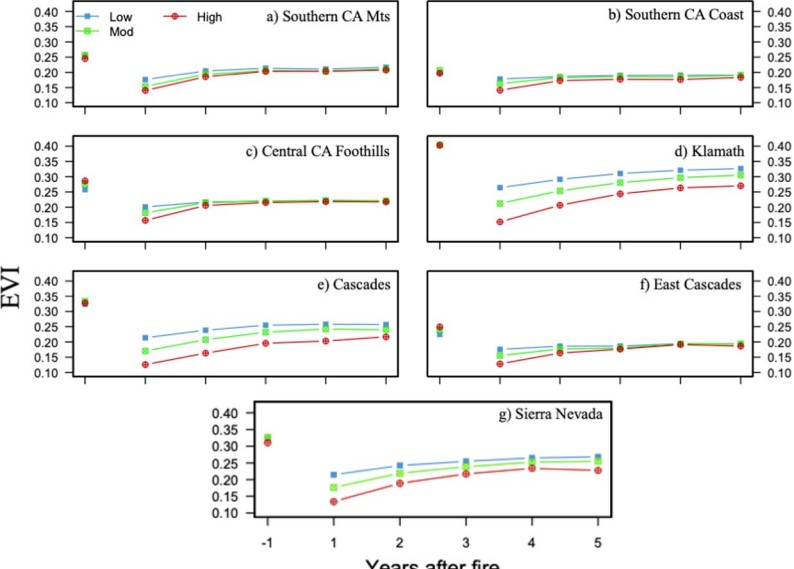

**Fig 5.** Prefire EVI and five-year early-summer postfire trajectory (a-g). Average early-summer postfire EVI for one-year pre-fire through five-years post-fire for each ecoregion SCM, SCC, CCF, K, C, EC, and SN. Prefire and postfire EVI are burned area averages.

wildfire-induced EVI change while the K ecoregion had the largest. In addition, K had the fastest rates of recovery for all burn severity classes relative to the prefire averages. The comparatively quick recovery of EVI towards a plateau in the SCM, SCC, and CCF (~1–3 years) may be associated with the more rapid recovery of their predominant grassland and savanna vegetation types, as these ecoregions experienced a large proportion of low and moderate severity burns (Fig 5B). Also, five-year average early-summer EVI after high severity fires remained lower than all other burn severity classes.

## 3.3 Albedo change

Average prefire albedo was highest in the SCM, SCC and CCF ecoregions (0.123–0.144) and lowest in the K, C, EC, and SN (0.101–0.124) (Fig 6A–6C). The East Cascade ecoregion had the largest variation in average prefire albedo among burn severity classes (0.01). Prefire albedo values were lowest in high severity burned areas in the SCM, SCC, CCF, C, and EC ecoregions.

In the first year after fire, albedo decreased below prefire levels in all three burn severity classes in the K, C, EC, and SN ecoregions, but increased above prefire levels in the SCM, SCC, and CCF. In the K, C, EC, and SN, albedo levels generally increased continuously for five years post-fire, and albedo values in the fifth-year post-fire were above prefire levels in the C, EC, and SN.

Postfire trajectories in the SCM, SCC, and CCF displayed different behavior relative to the northern California ecoregions. Whereas the K, C, EC, and SN ecoregions experienced one-year postfire decreases, albedo values increased up to 0.02 in the SCM and 0.01 in the SCC after the first year, with the largest increases in the most severely burned areas (Fig 6A and 6B). However, albedo in the CCF did not change substantially in any burn severity class after one year, but both the moderate and high burn severity class had an increase between year one and two (Fig 6C). In the ecoregions that experienced decreases after the first-year post-fire, albedo values tend to exceed prefire values by years two and three (Fig 6D–6G).

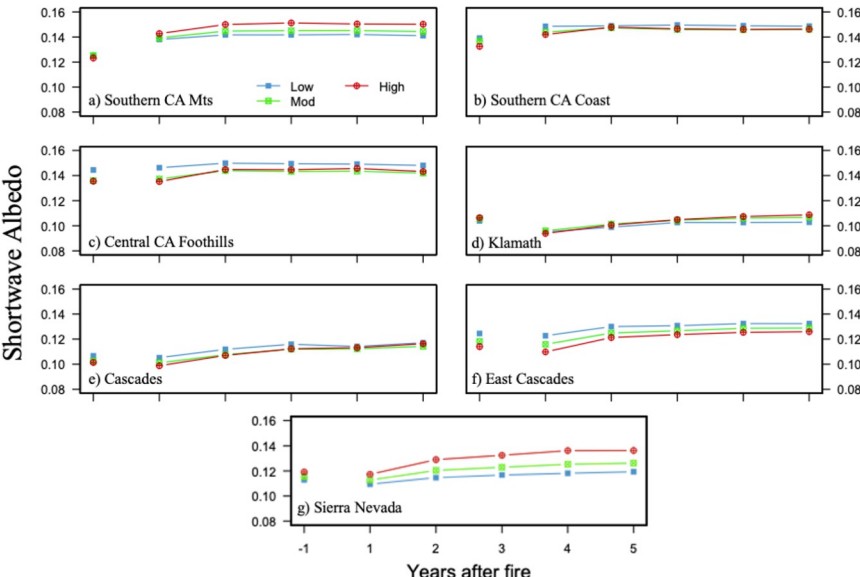

**Fig 6.** Prefire albedo and five-year early-summer postfire trajectory (a-g). Average early-summer surface shortwave albedo response for the first-year pre-fire through five-years post-fire for each ecoregion SCM, SCC, CCF, K, C, EC, and SN. Prefire and postfire albedo are burned area averages.

### 3.4 Land surface temperature

Average prefire LST is lowest in the K ecoregion (305.4 K) and highest in the SCC (316.68 K) (Fig 7 and Table 2). The greatest prefire LST difference among burn severity classes within a single ecoregion is 1.9 K in the CCF. However, prefire LST in regions that experienced high severity fires was higher than it was in regions of low and moderate severity in K, SN, SCM,

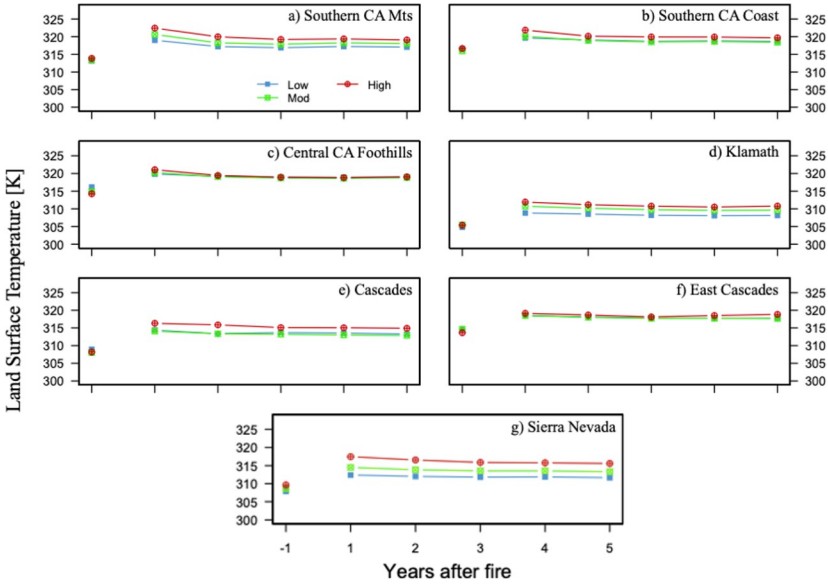

**Fig 7.** Prefire land surface temperature and five-year early-summer postfire trajectory (a-g). Average early-summer land surface temperature first-year pre-fire through five-years post-fire for each ecoregion SCM, SCC, CCF, K, C, EC, and SN. Prefire and postfire LST are burned area averages.

and SCC ecoregions. LST increased in the first-year post-fire in all ecoregions regardless of burn severity, and the greatest increases in LST were associated with high severity fires (Fig 7). Increases in LST after low severity fires ranged from 3.1 in the CCF to 5.8 K in the SCM, while increases in LST after high severity fires ranged from 5.5 K in the EC to 8.6 K in the SCM.

The rate at which LST recovered towards prefire levels was different for each ecoregion, however, overall, postfire LST did not return to prefire temperatures after five years in any ecoregion in any burn severity class. Furthermore, LST after high severity fires rose to a higher temperature and remained elevated over temperatures in low and moderate burn severity areas in the SCM, SCC, K, C, and SN. Across all ecoregions and burn severity classes, the trajectory of postfire temperature over five years tended to remain stable, with minimal increases and decreases in temperature after the initial postfire increase in the first year.

## 4 Discussion

### 4.1 Role of burn severity in vegetation recovery

We found that three features describe most of the relationship between vegetation recovery response and burn severity in California's ecoregions. First, the largest decreases in first year postfire EVI, regardless of burn severity class, occur in the same ecoregions that have the highest mean prefire early-summer EVI (i.e. K, C, and SN) (Table 1 and Fig 5). This pattern indicates that all fires, regardless of burn severity level, tend to result in greater amounts of vegetation removal in ecoregions with high prefire EVI (e.g. high biomass areas). For example, the average prefire EVI value for low severity fires in the Klamath ecoregion is 0.4, and the average ΔEVI one-year post-fire is 0.14. So, even though ΔEVI is greatest for high severity fires across ecoregions, the SCC ecoregion, with one of the lowest average prefire EVI values for high severity fires (0.2) experiences a 0.06 one-year postfire decrease–less than half of the Klamath ecoregion's low severity ΔEVI. Our results that show strong correlations between RdNBR and ΔEVI, as well as large prefire EVI in northern California ecoregions that experience the greatest first-year ΔEVI, are consistent with studies that found that higher prefire EVI in the Cascade and Klamath regions of northern California were associated with a greater likelihood of high severity fires [54–56]. Furthermore, [57] found that low severity fires were more likely to occur in areas with lower prefire EVI, which is similar to what we found in the spatial distribution of RdNBR derived burned area in southern California (Fig 2B). In addition, the impact of wildfire on NDVI, a similar vegetation index as EVI, was greater for needleleaf trees than it was for shrubs, an effect that was corroborated with satellite imagery and aerial photos [58].

Second, fires that burned at high severity are associated with greater ΔEVI across all ecoregions compared to less severe fires (Fig 5). In other words, changes in the amount of live fuel between the prefire and one-year postfire images increases with burn severity. This relationship, demonstrated by the strong negative correlation between ΔEVI and RdNBR (i.e. large changes in EVI after higher severity fire), is expected, as both indices are primarily influenced by the magnitude of change between near and shortwave infrared wavelengths. However, the high correlation observed between ΔEVI and RdNBR may also be a result of the previously mentioned association between vegetation type and degree of EVI or NDVI change [54].

The third feature that describes the relationship between postfire vegetation recovery and burn severity is the rate at which EVI approaches prefire levels over the course of five years. Throughout the five-year postfire interval, relative to average prefire EVI values, EVI recovers faster after high severity fires in all ecoregions. Rapid recovery of vegetation in the first years after wildfire is likely due to the growth of shrubs and other herbaceous vegetation. Furthermore, our results are corroborated by other studies that show that, while NDVI (a vegetation

index similar to EVI) recovers fastest during the first two to three years post-fire, the effects of high severity burns can still be observed five or more years later, and lower NDVI values were found after high severity fires [22, 59, 60]. Further, in the ecoregions that had smaller ΔEVI after the first year (i.e. SCM, SCC, CCF, EC), EVI trajectories for low, moderate, and high burn severity tended to converge by the third-year post-fire (Fig 5A–5C and 5F). By identifying which ecoregions are more likely to experience high severity burns, land and fire management can focus resources on areas experiencing high degrees of postfire ecological change and erosion, or are at an increased risk for habitat endangerment [17, 51].

## 4.2 Burn severity and albedo change

One of the most immediate effects of wildfire on the land surface is the removal of vegetation and the deposition of ash [8, 61]. By definition, high severity fires are often stand replacing disturbances that lead to rapid changes in the biophysical characteristics of the land surface. Abrupt changes to the partitioning of energy from latent to sensible heat flux (associated with the removal of vegetation) can increase the amount of net radiation available at the surface [8, 62].

Postfire albedo change may be highly dependent on plant type, phenology, and burned severity, as a predominantly forested region will likely experience a different postfire albedo trajectory than a grassland or savanna (Fig 6C and 6D). Postfire albedo may decrease after high severity fire in a biomass rich area like the K, C, or SN ecoregion for a year or more because forest recovers at a slower rate than grassland and shrub (Fig 6D, 6E and 6G). On the other hand, ecoregions that contain predominantly grassland and savanna land cover types (i.e. SCM, SCC) are more likely to experience initial increases in postfire albedo, as one year is a sufficient amount of time to remove all traces of ash deposited on the surface and for vegetation to start recovering (Fig 6A and 6B). It is important to note that ash residence time varies widely, and decreases in albedo caused by ash deposition can be offset by the regrowth of early successional plants like shrubs and grasses, which tend to have higher albedos [20, 30, 34, 63]. In fact, one study found that albedo increases in the first postfire summer are likely related to exposure of bare ground after ash is dispersed in the winter, while subsequent increases each year following are due to the regeneration of vegetation [10]. This effect was observed within the SCM, SCC, and CCF ecoregions, where albedo in the first-year post-fire increased, while the K, C, EC, and SN ecoregions experienced decreases (Fig 6).

Prefire albedo levels were lower, and fires of all severity levels resulted in greater decreases in the K, C, and SN ecoregions relative to the SCM, SCC, CCF, and EC, likely due to the differential effects of fire on vegetation with higher EVI values. In addition, ecoregions with an abundance of live fuel (i.e. high EVI), tend to have lower albedos. Thus, greater average prefire EVI and lower average pre-fire albedo are both associated with high severity fires. Additionally, seasonal differences in albedo may also be responsible for some of the variation in first-year postfire values, as soil moisture content and plant growth during and after the rainy season has a direct effect on albedo [10, 35].

## 4.3 Burn severity and land surface temperature

To our knowledge, a small number of studies have investigated the connections between burn severity and land surface temperature, and only a few for regions within the state of California [10, 64–66]. Further investigation into the relationships between burn severity, vegetation recovery, and LST may lead to the use of LST as indicators of burn severity, as postfire increases in temperature were found to change proportionally with NDVI and slowly return to pre-fire levels as vegetation recovers [64]. Additionally, the analysis of postfire recovery

patterns of land surface temperature will strengthen our understanding of the impacts of fire on surface energy balance and improve the reliability of simulated land surface processes within models.

Albedo and land surface temperature are closely related biophysical characteristics of the land surface that are impacted by wildfire and burn severity [50, 66]. Wildfire-induced alterations to the land surface can persist for years, leading to increased aridity, as well as changes to diurnal energy and temperature fluctuations. In fact, there may be significant seasonal variation in the wildfire related impact on land surface temperature, with large changes during the summer and small changes during the winter [10]. However, the initial decrease in albedo and increases in radiation absorption and surface temperature associated with post-fire ash deposition tends to be short-lived, as the gradual process of plant regeneration contributes to the return of land surface temperatures to their pre-disturbance levels.

Here, the relationships between ΔEVI, ΔLST, and burn severity are less clear, as LST increased post-fire in all ecoregions, but remains elevated for the duration of the postfire years, regardless of the amount of EVI recovery (Figs 5 and 7). For example, the K, C, and SN had the greatest ΔEVI in the first-year post-fire, as well as the fastest rate of recovery towards prefire levels, however, increases in LST in these ecoregions remain mostly stable, only gradually diminishing towards average prefire LST over five years. A study extending further than five years post-fire would be necessary in order to investigate the length of postfire temperature change in California ecoregions.

The projected impacts of climate change on California ecosystems, including increasing temperatures and vapor pressure deficit, may significantly alter existing fire regimes [67, 68]. In general, warmer temperatures and drier conditions increase the risk for wildfire (assuming the presence of vegetation), and the addition of further postfire increases in temperature may push already stressed ecosystems past their ability to adapt [69].

## 5 Conclusions

We analyzed the influence of burn severity on vegetation recovery, albedo, and land surface temperature in seven California ecoregions between the years 2003 and 2020 based on MODIS satellite data derived products. Normalized Burn Ratio datasets were used to calculate RdNBR for the entire state for the duration of the study period. Early-summer averages were stratified into three burn severity classes and their prefire average and trajectory for the first five years post-fire plotted. Strong negative correlations were found between postfire NBR and RdNBR, as well as ΔEVI and RdNBR. We found that the largest decreases in EVI after one-year post-fire occur in the ecoregions with the highest prefire EVI values (i.e. K, C, SN). In addition, the greatest decrease in EVI and the fastest recovery towards prefire values occurred after high severity fires (in all ecoregions). Also, EVI did not recover to prefire values after five-years post-fire in any burn severity class in any ecoregion. We found that the lowest prefire albedo values occurred in the same ecoregions with the highest prefire EVI. First-year postfire albedo decreased in the K, C, EC, and SN ecoregions, but increased after one year in the SCM, SCC, and CCF. Differences in first-year postfire change are likely due to variations in the duration of ash residence times, rate of plant regeneration, and average albedo of early successional plants (such as grasses and shrubs). After five years post-fire, albedo values were larger than prefire values in all ecoregions except K. We found that first-year postfire increases in LST were greatest within high severity burned areas. All ecoregions experienced a postfire increase in LST that remained relatively stable throughout the five-years post-fire with only gradual decreases towards prefire levels.

An improved understanding of the biophysical response to large-scale wildfires becomes increasingly important as California's high summertime temperatures and seasonal summer

droughts continue to drive record-breaking fire seasons. Furthermore, the spatial distribution of burn severity, along with a comprehensive understanding of postfire response of important indicators of ecosystem health (like the presence of vegetation and large changes in LST), can inform fire and land management in their efforts to effectively mitigate and suppress large wildfires in key areas. In addition, the temporal analysis of postfire EVI, albedo, and temperature may improve the accuracy and inform the conceptualization of future modeling of wildfire impacts on land surface processes.

## Acknowledgments

The authors would like to thank the SDSU Center for Earth System Analysis Research for providing the computer facilities necessary for the completion of the study.

## Author Contributions

**Conceptualization:** David E. Rother.

**Data curation:** David E. Rother.

**Formal analysis:** David E. Rother, Doug Stow.

**Investigation:** David E. Rother.

**Methodology:** David E. Rother.

**Software:** David E. Rother.

**Supervision:** Fernando De Sales, Joe McFadden.

**Visualization:** David E. Rother.

**Writing – original draft:** David E. Rother.

**Writing – review & editing:** David E. Rother, Fernando De Sales, Doug Stow, Joe McFadden.

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
