## [Decision Letter · Decision Letter 0]

22 Jun 2022

PONE-D-22-09676Impacts of burn severity on short-term postfire vegetation recovery, surface albedo, and land surface temperature in California ecoregionsPLOS ONE

Dear Dr. Rother,

Thank you for submitting your manuscript to PLOS ONE. After careful consideration, we feel that it has merit but does not fully meet PLOS ONE’s publication criteria as it currently stands. Therefore, we invite you to submit a revised version of the manuscript that addresses the points raised during the review process.

Reviewer 1 has indicated that major changes are required before publication and has provided thorough comments. Reviewer 2 has suggested the manuscript be rejected on the basis of lacking novelty, but this is not how manuscripts are assessed for PLOS ONE.

We look forward to receiving your revised manuscript.

Kind regards,

Paul Pickell, Ph.D.

Academic Editor

PLOS ONE

Journal Requirements:

3. We note that Figures 1 and 2 in your submission contain map images which may be copyrighted. All PLOS content is published under the Creative Commons Attribution License (CC BY 4.0), which means that the manuscript, images, and Supporting Information files will be freely available online, and any third party is permitted to access, download, copy, distribute, and use these materials in any way, even commercially, with proper attribution. For these reasons, we cannot publish previously copyrighted maps or satellite images created using proprietary data, such as Google software (Google Maps, Street View, and Earth). For more information, see our copyright guidelines: http://journals.plos.org/plosone/s/licenses-and-copyright.

 a. You may seek permission from the original copyright holder of Figure(s) [#] to publish the content specifically under the CC BY 4.0 license. 

Reviewers' comments:

Reviewer's Responses to Questions

**Comments to the Author**

1. Is the manuscript technically sound, and do the data support the conclusions?

Reviewer #1: Yes

Reviewer #2: No

2. Has the statistical analysis been performed appropriately and rigorously? 

Reviewer #1: Yes

Reviewer #2: No

3. Have the authors made all data underlying the findings in their manuscript fully available?

Reviewer #1: Yes

Reviewer #2: No

4. Is the manuscript presented in an intelligible fashion and written in standard English?

Reviewer #1: Yes

Reviewer #2: Yes

5. Review Comments to the Author

Reviewer #1: I'd like t congratulate the authors in getting this work submitted and into review. It is great, timely, and really nice use of the MODIS products. I enjoyed reading and reviewing the paper and it is clear you all put a lot of effort into making it accessible to the readers, researchers, and land managers that will be interested in it. Thank you for your efforts. I have compiled a few spots where the text will need some clarification listed below by line number - I believe most of these are just areas where the text could use some additional words to help it along or the methods you are using need some supplemental justification.

Line 66-> "The growth" suggest changing it to this or that to indicate you referring to the growth in burned area from the previous sentence.

line 99 -> While I agree that a comprehensive understanding of the biophysical response to wildfire and detailed spatial burn severity patterns are required, I do not think this paper will report on everything that is considered comprehensive in this case. As my reading of this paper, you do report on Temperature, EVI, RDNBR (and its components) measured from MODIS data. Some would argue that MODIS does not provide the detail necessary, and that these things that you are reporting on are not considered comprehensive enough (what about soils, more than 5 years of data after fire, more detailed than 500m data is available).

Line 111: State should be lowercase.

Line 112: Critical biophysical variables: please indicate which ones. This is your "I'm going to do this with that and want to investigate this thing" paragraph; you should tell us this info.

line 140-143 AND Fig 1 caption: The reasons do not match. Please pick one or both and go with it.

Table 1: Other ecoregions were excluded for low burn amounts, but Cascades experienced only 6% burn, which is remarkably low. Why was it also kept, if indeed that amount of burned area was that low?

line 165: I would ask about the use of predominant when referring to TWO landcover classes, as predominant in a singular - I would suggest prevailing or most substantial.

line 181: though it is indicated that EVI can distinguish between canopy and canopy background, it is not clear how that particularly matters in the case of burns that you are interested in. In a high severe fire, canopy, ground, and anything in between is burned. In a low severity fire, one would say maybe some canopy trees may die out due to fire, but the majority of the fire is at the ground level clearing out ground level vegetation. So, if this version of EVI is only reporting on canopy, then it would be quite problematic four your later analysis where you partition fires in severity classes based on cumulative distributions.

line 207: NBR at each month, but, and this comment will surface again, when was this data used? I understand early summer NBR was used to calculate a number of items, but the later summer? Fall? Winter? Spring? There is no mention of those months data from here out. And, of course, apologies to the authors if I missed where they are used.

line 213: LOW NBR values could also just indicate low vegetation and not just recently burned areas - it is a possibility worth considering.

Line 214: Why was the decision made to look at fires at a homogenous blob - an area and not in a per pixel method. at 500, pixels, there can be a fair amount of variability, and this is ok., but when considering that fires have a growing trend over this time period under observation, you could be averaging something else when aggregating a large fire to an average. This question also occurred to me later in the manuscript, when discussing the spatial pattern. It registered with me that large fires occur over months sometimes, and over large areas, potentially burning through many varied terrains and vegetation assemblages. Also, this variation would potentially leave differing levels of burn severity and patterns. This seem to be brushed aside for a simple average. While I understand there are many ways to analyze these data, I would like to know (and likely other readers) why the decisions was made to simplify these fires to single averages.

Line 214 part 2: Similarly, why report the area and number if pixels burnt in fires in the ecoregion back in Table 1, if they will be compressed down to an average. If compressed, your regressions for correlations will be on the averages not the pixels. Thus it would be helpful to know the n of the fires and timing of those fires instead of number of pixels under observation.

Line 252: Again I am asking for some clarification - where were the monthly data used? It seems as if the obtained averages EVI and other variables from burn perimeters once a year for five years. Please clarify

Line 253: not related to this line, but important to note here: were fires in 2020 considered? what was the last year in the time series to consider fires under observation. if 5 years post fire data is needed, then 2015 must be the last year fires can occur. This is not stated in the manuscript, and could be stated for clarification.

Table 2: this is difficult to step through - and readers typically see a wall of numbers and skip over it. BUT I would suggest

that there are some important points here. Where did the +/- values come from? Could this be better shown as a figure? It seems as if the X axis is missing its label. Again, number fires may be more helpful here too, as your unit of observation is an average within a fire perimeter.

Line 304: Are there any years that are outliers? Perhaps very hot and dry years where fires grew so large and so hot burning throughout the season?

line 322: reference to a plot that is not present in the text. Suggest adding see Figure 4a here.

Line 324: how were these thresholds decided? At what point in the cumulative distribution - was it % ? was it an average? it is not stated in the text.

Line 324: Another thought here: the time series is almost 20 years long, but how is the effect of time on this series? what id fires are larger and more severe at the tail end of the time series?

line 344: Some ecoregions are prone to more severe burns, some less so. how is this understood in this research?

line 344: The manuscript is mainly concerned with disturbance, but recovery is in the title. VERY little is actually said about recovery, and it is not exaclty disucssed what recovery means in this context. Vegetatively, ecologically, even numerically using the NBR/EVI/DNBR data. Please make it clear what is considered recovery.

Line 167: going back to landcover decisions in table 1 - why were the "closed shrublands and woody savanna vegetation" combined into one category?" If there were distinct enough for the MODIS Science Team to classify them, should not be separate. In my mind, these are two very different land covers, with two different fire regimes.

Line 358: Is correlating RdNBR with post fire NBR - it seems all too circular. post fire NBR is an essential component to dNBR, and thus RdNBR, so it make sense that they are correlated - they are expressing similar things. It is the further claim that post fire NBR is a powerful control on RdNBR that is unsupported. My analogy here is to cake. If you made a cake, you would find that flour is correlated with cake. it is not a surprise, and flour is not so much a control on cake as it is an ingredient. These lines feel circumlocutory. Please either remove or clarify or justify

Table 4: the X axis appears un labeled. Also, what is being regressed here -> severity classes, landcovers, entire ecoregions? the text does not make this clear.

line 374-397: Twenty three lines about recovery, though what recovery means is not defined. The manuscript implies that a return to pre fire levels of EVI is expected in five years is not met in any ecoregion. And that should be expected considering some of the ecoregions covered are needle leaf forests.

line 418: significantly implies a test, and a p value. if this is not the case, I suggest using the word substantial instead.

line 426: temperature here - change of 1.9K means almost 1.9 C, which is regarded as "not varying significantly (again suggest substantial)" but 1.5 degrees of warming in climate change literature is considered almost catastrophic. If leaning into using climate change (as is done in the conclusion) as a driver of fire severity and size, maybe put this in context better.

line 437: the regressions - it is not clear again if they were completed on pixels or burn perimeter averages - or burn perimeter averages within landcover within an ecozone. please clarify the unit of observation

line 459: the pacific north west is a region, but not contained in this study area. and the ecoregion in California was excluded as well. please rephrase for clarity - suggest just saying Klamath ecoregion.

line 461: pattern and spatial distribution are not the same thing - pattern implies an analysis was completed on the spatial distribution, but the manuscript does not show that. I suggest omitting the word pattern here.

Line 480: I can understand referring to NDVI, but there must be some research on fire that examines EVI also. The switch to NDVI at this point does not help the reader to understand what is happening.

Line 481: suggest using a different word for furthermore - perhaps moreover. or further.

Line 489: Stand Replacing. Not all ecosystems under observations are forests in this manuscript. I do not suggest using this forest centric wording.

Line 521-523: I am unsure where the analysis on NDVI was performed in this paper. Is there a missing reference to another paper here?

line 546: yes, i strongly agree with this statement.

figure captions: in general these need more description, as they cannot stand on their own.

Again, thank you for your efforts and this research. It is very interesting exploration of fire in CA using MODIS data products.

Reviewer #2: This study used the MODIS-derived RdNBR dataset to analyze the impact of burn severity on the five-year postfire early-summer averages of EVI, albedo, and land surface temperature between the years 2003 – 2020. Generally, the paper is well organized and has good writing. However, the analysis of different changes of these parameters to the burn severity after the fire has already been discussed in the past studies, and it may not have a good fit for the journal scope in terms of lacking novelty. On the other hand, the spatial resolution of MODIS is too coarse to detect small-scale fires, resulting in the lacking of analysis of small-scale fires with relatively lower burn severity. Therefore, the overall merit of this study is low, and I cannot recommend publishing it in Plus One.

6. PLOS authors have the option to publish the peer review history of their article (what does this mean?). If published, this will include your full peer review and any attached files.

Reviewer #1: No

Reviewer #2: No

---

## [Author Response · Author response to Decision Letter 0]

12 Jul 2022

We would like to extend our appreciation to the reviewer for the constructive comments and recommendations toward improving our manuscript. Your acknowledgement of the hard work and effort that went into the data collection, processing, and analysis, as well as the writing of the manuscript is greatly appreciated. All issues raised by the reviewer have been addressed and the manuscript has been updated accordingly. The following are our point-by-point responses to the reviewer’s comments:

Line by line comments [Line references indicate the location of the revised text and not the reviewer’s initial line]:

Line 66 -> changed in text, edited for clarity

Line 102-103 -> removed “comprehensive” as a descriptor and replaced it with “ecoregion-level”

Line 114 -> changed “State” to “state”

Line 115-116 -> removed “critical biophysical variables” and replaced it with the actual variables, “EVI, land surface albedo, and temperature”

Line 161-169 -> Added information to the caption in Figure 1 for clarity. It now more closely aligns with what is stated in the previous paragraph (143-147)

Table 1 -> In order to address the concern about the Cascades ecoregion only experiencing 6% burned area but still being included in the study, we added the total burned pixels and the percent burned area for the Marine West Coast Mountains and the North American Desert ecoregions to the text for reference (lines 143-147)

Line 188-191 -> rephrased the sentence without the use of “predominantly”, used suggested “prevailing” and “most substantial” instead

Line 210-212 -> Here we address the comment about EVI and its ability to differentiate between canopy and canopy background. EVI does not only report on canopy as the it says in the Reviewer comment. One of the main positive attributes of the data is its ability to distinguish the canopy from the canopy background. This is ideal for investigating the trajectory of EVI after fires of all severities, as changes to understory vegetation after low severity fires will still show up in the EVI averages.

Line 239-240 -> We clarified what time period the NBR was being calculated for in each year of the study period. The same early summer period (June 26 -August 12) was used for all biophysical variables in the study

Line 246 -> used the Reviewer comment to add a qualifying statement that low NBR may also represent low vegetation

Line 247 -> We made the decision to study the changes occurring within burned perimeters as areal averages instead of within pixels because we wanted to focus on the temporal aspect of biophysical change after wildfire. We are aware that some of the spatial heterogeneity of burn severity within fire perimeters was sacrificed when using MODIS data. However, MODIS has high temporal frequency and wide area coverage which makes it perfect for a temporal analysis of how EVI, temperature, and albedo change within burn perimeters throughout the five years after wildfire. 

 For our methodology, we are using the sum of all burned area for each year in our analysis. For example, 2003’s burned perimeters are used to get early-summer averages of EVI, albedo, and temperature in 2002 (prefire), and then in 2004, 2005, 2006, 2007, and 2008. This does away with the issue of growing wildfires over time in terms of total burned area because each year’s total burned area is considered. We also acknowledge that by using prefire NBR and postfire NBR that are taken a year apart, we are factoring in a year of recovery into the analysis of burn severity, EVI, albedo, and temperature. We believe that by using this methodology we can quantify the largest changes between a burned and unburned land surface. We plan on future studies where we investigate the spatial patterns within burn perimeters. 

Line 247 Part 2 -> The area and number of burned pixels shown in Table 1 and 2 are meant to clue the readers into the general occurrence of wildfire in each ecoregion. These stats are meant to give the reader a sense of how much land has been burned in each ecoregion, as well as which ecoregions tend to burn the most. We are less concerned with the timing and number of fires that burn each year than we are with the mean climatology of the biophysical variables in the fire perimeters. Providing a detailed list of the number and timing of every fire that burned in California between 2003-2020 is outside the scope of the study. We are not trying to confuse the reviewer or the reader into thinking that the correlations shown later in the paper are calculated spatially, these correlations are calculated using areal averages within burned perimeters in the 7 different ecoregions. 

Line 290 -> Early summer averages are an average for June 26 – August 12 for that variable within all the burned area within a given ecoregion. We take the average within the burned area one year prior and one to five years postfire (all the same burned area). Then all the prefire averages for that ecoregion are averaged, then all the one year, 2 year, etc postfire averages are averaged to get a single time series for each ecoregion for each variable (which are Figures 5, 6, 7). The Reviewer is correct in saying that EVI, albedo, and temperature averages are obtained by taking the average of each variable within a given years burned perimeter once a year for 5 years. See Line 247’s comment for more detail about the methodology. 

Line 290 -> Fires in 2020 were considered – 2020 was the last year in which burned area perimeters were considered. We did not require that 5 years of postfire data be available in order for us to include the years 2016-2020. That means that 2016 includes 5 years of postfire data (because we use MODIS data for postfire analysis through 2021, but not burned area data in 2021), 2017 includes 4 years of postfire data, and 2018 includes 3 years, etc. Our study’s findings are limited by the satellite data temporal availability (2002-2021), and future changes in fire and climate regime associated with global warming may impact our findings. In order for us to consider the greatest number of years possible, we chose to use burned area data through 2020, even though it meant only considering one year post-fire for that year.

- We added a clarifying statement addressing this issue to the paper in Line 293-295

Table 2 -> The +/- values are the standard deviation of the mean values within ecoregions. I added this information to the Table 2 legend. We appreciate the reviewers comment about including the number of fires, however, we believe that because we are taking an average of each variable within burn perimeters/ecoregions, the total number of burned pixels from which these means/standard deviations are calculated is more beneficial to the reader. 

Line 331 -> Reviewer may refer to Figure 3 and Lines 371-379 (and 361-364) for information about which years in the study area burned the most. Years 2018 and 2020 were certainly outliers in terms of total area burned in California. An investigative look at the annual summertime temperatures and humidity within each ecoregion is outside the scope of the project. However, some future work of the corresponding author does plan to look at both historical and projected fire weather within these same ecoregions. 

Line 382 -> added Figure 4 to the text to address the problem of referring to something that did not exist. 

Line 380 -> Refer to previous Lines 266-274 (methods section) for how the burn severity threshold were derived. 

Line 382 -> The reviewer is correct in stating that the size of the fire does not affect the postfire averages because the time series is calculated by taking the average of each variable within the burned perimeter. Because all the prefire years, one year, two year, etc points are averaged together to obtain the final time series, the size of the fire (regardless of when the fire occurred – at the beginning or end of the time series) will not affect the final averages (it will just be a product of more points being averaged together, relative to other ecoregions that may have experienced fewer burn points). 

Line 402 -> The methodology for obtaining the burn severity thresholds (Figure 4) uses the cumulative distribution of RdNBR values within pixels. These thresholds are defined based on the total number of burned pixels within each ecoregion, and not on the size of the ecoregion itself. For example, Figure 4 shows the reader that even though Klamath is not the second largest ecoregion, it has the second most occurrence of moderate to high severity burn pixels. In this way the reader can see that the Klamath and the Sierra Nevada are prone to higher severity burns, even though they are not necessarily the largest ecoregions. 

Line 402 -> Thank you for pointing out that a definition of vegetation recovery was missing from the paper. We added a definition to the methods section where we describe the EVI datasets (Lines 212-214)

Table 1 -> We made the decision to combine the two land cover categories because they contain a similar “fractional coverage of woody vegetation”. This reasoning was also found in:

Jin Y, Randerson JT, Goetz SJ, Beck PSA, Loranty MM, Goulden ML. The influence of burn severity on postfire vegetation recovery and albedo change during early succession in North American boreal forests. Journal of Geophysical Research: Biogeosciences,. 2012; 117, G01036. https://doi.org/10.1029/2011JG001886.

Line 417 -> On line 421 we state that high correlations between RdNBR and EVI are expected because they are both sensitive to infrared reflectance. We have opted to follow the reviewer’s suggestion and remove the lines about how NBR is a control on RdNBR. Thank you. 

Table 4 -> We have followed the Reviewer’s suggestions and labeled the x axis, added information to the caption and specified what is being regressed in the table. 

Line 456 -> We added what is meant by vegetation recovery to the methods section after reading a previous comment. We removed part of a sentence that says that EVI return to prefire values was not found in any ecoregion “which is to be expected after such a brief interval”. We agree with the reviewer that this could be misleading to the reader in its implications. 

Line 493 -> We have changed the wording from “significantly” to “substantially” following the reviewer’s suggestion

Line 507 -> We decided to remove the wording of “does not vary significantly” entirely to avoid confusion about the magnitude of that 1.9K change in prefire LST

Line 520 -> It is unclear which regressions the reviewer is referring to at this point in the text, but the answer to their question is that average LST was calculated within burned perimeters within each ecoregion. Land cover averages were only calculated for the initial tables and are not part of the later analysis, apart from the RdNBR cumulative frequency distributions (Results)

Line 547 -> changed the wording, removed Pacific Northwest and replaced with Klamath as suggested

Line 549 -> removed the word “pattern” as suggested

Line 572 -> added a statement that helps the reader understand what NDVI is in this context

Line 575 -> replaced “Furthermore” with “Further” as suggested

Line 583-584 -> foundational literature including Miller and Thode (2007) refer to complete mortality (high severity fire) of all live vegetation as “stand replacing”, regardless of type. This sentence is informing the reader on literature and is not necessarily about our findings, and we do not use the wording here to imply a forest ecosystem. 

Line 620 -> added a citation, 64

Line 646 -> We appreciate that, and all of your comments. Thank you so much for your time.

Figure captions -> we went through the text and added information to figure captions as recommended by the reviewer

---

## [Decision Letter · Decision Letter 1]

30 Aug 2022

Impacts of burn severity on short-term postfire vegetation recovery, surface albedo, and land surface temperature in California ecoregions

PONE-D-22-09676R1

Dear Dr. Rother,

We’re pleased to inform you that your manuscript has been judged scientifically suitable for publication and will be formally accepted for publication once it meets all outstanding technical requirements.

Kind regards,

Paul Pickell, Ph.D.

Academic Editor

PLOS ONE

Additional Editor Comments (optional):

Reviewers' comments:

Reviewer's Responses to Questions

**Comments to the Author**

1. If the authors have adequately addressed your comments raised in a previous round of review and you feel that this manuscript is now acceptable for publication, you may indicate that here to bypass the “Comments to the Author” section, enter your conflict of interest statement in the “Confidential to Editor” section, and submit your "Accept" recommendation.

Reviewer #1: All comments have been addressed

2. Is the manuscript technically sound, and do the data support the conclusions?

Reviewer #1: Yes

3. Has the statistical analysis been performed appropriately and rigorously? 

Reviewer #1: Yes

4. Have the authors made all data underlying the findings in their manuscript fully available?

Reviewer #1: Yes

5. Is the manuscript presented in an intelligible fashion and written in standard English?

Reviewer #1: Yes

6. Review Comments to the Author

Reviewer #1: Thank you for making the suggested changes. The depth you've added helps the reader understand what you did and puts the results in better context. nice work!

7. PLOS authors have the option to publish the peer review history of their article (what does this mean?). If published, this will include your full peer review and any attached files.

Reviewer #1: No

---

## [Editor Report · Acceptance letter]

12 Oct 2022

PONE-D-22-09676R1 

Impacts of burn severity on short-term postfire vegetation recovery, surface albedo, and land surface temperature in California ecoregions 

Dear Dr. Rother:

I'm pleased to inform you that your manuscript has been deemed suitable for publication in PLOS ONE. Congratulations! Your manuscript is now with our production department. 

Kind regards, 

on behalf of

Dr. Paul Pickell 

Academic Editor

PLOS ONE